# Next Generation Cereal Crop Yield Enhancement: From Knowledge of Inflorescence Development to Practical Engineering by Genome Editing

**DOI:** 10.3390/ijms22105167

**Published:** 2021-05-13

**Authors:** Lei Liu, Penelope L. Lindsay, David Jackson

**Affiliations:** Cold Spring Harbor Laboratory, Cold Spring Harbor, NY 11724, USA; lliu@cshl.edu (L.L.); lindsay@cshl.edu (P.L.L.)

**Keywords:** crop-yield improvement, breeding, inflorescence development, genome editing

## Abstract

Artificial domestication and improvement of the majority of crops began approximately 10,000 years ago, in different parts of the world, to achieve high productivity, good quality, and widespread adaptability. It was initiated from a phenotype-based selection by local farmers and developed to current biotechnology-based breeding to feed over 7 billion people. For most cereal crops, yield relates to grain production, which could be enhanced by increasing grain number and weight. Grain number is typically determined during inflorescence development. Many mutants and genes for inflorescence development have already been characterized in cereal crops. Therefore, optimization of such genes could fine-tune yield-related traits, such as grain number. With the rapidly advancing genome-editing technologies and understanding of yield-related traits, knowledge-driven breeding by design is becoming a reality. This review introduces knowledge about inflorescence yield-related traits in cereal crops, focusing on rice, maize, and wheat. Next, emerging genome-editing technologies and recent studies that apply this technology to engineer crop yield improvement by targeting inflorescence development are reviewed. These approaches promise to usher in a new era of breeding practice.

## 1. Introduction

The global population is expected to increase to 9.2 billion in 2050, and agricultural production needs to increase by about 70 percent from current levels to meet the increased food demand, as predicted by Food and Agriculture Organization (http://www.fao.org/wsfs/forum2050/wsfs-background-documents/issues-briefs/en/, accessed on 1 February 2021). Cereal crops, such as rice, wheat, and maize, are the world’s most important sources of calories for humans, livestock feed for animals, and raw material for biofuel [1]. However, with the threat of urbanization, land erosion, sea-level rise, and pollution, the arable land for cereal-crop production will become more limited [2]. Therefore, improving cereal-crop-grain production is critical to meet further demand.

A majority of modern crop varieties were domesticated from their wild ancestors within the past ~12,000 years [3]. During domestication, plants were selected to render them easier to breed, culture, harvest, and store seeds [4]. In the past century, domesticated crops underwent improvement to achieve high productivity and widespread adaptability, by pyramiding beneficial mutations and recombinants in key genes [5]. Adopting new technologies, such as hybrid breeding, high-yielding dwarf wheat and rice varieties, and genetic modification by transformation, the grain yield of cereal crops has risen steadily [4,5,6]. Nowadays, rice, wheat, and maize supply nearly half of the calories consumed by humans, suggesting that their production is critical to feed an increasing population [1].

Dissecting the genetic changes during crop domestication and improvement is critical to understand the mechanistic basis of grain yield, and to guide breeding efforts towards developing high-yielding varieties [1]. Grain yield of cereal crops is a complex trait controlled by numerous quantitative trait loci (QTL). Current crop yield enhancement relies heavily on natural genetic changes [4,5]. Great progress has been made in mapping and cloning these yield-related QTLs in crops [7,8,9,10,11,12,13,14]. However, a majority of natural variants underlying yield-related traits have a minor effect, and those that have a greater effect may act only in specific genetic backgrounds [7,8,9,10,11,12,13,14], challenging their application in yield enhancement. On the other hand, climate change is spurring extreme environmental conditions, including drought, heat, cold, saline, and alkaline soils [2]. Therefore, geneticists and breeders face the challenge of enhancing the yield of cereal crops through genetic improvement of germplasm to bridge the gap between production and demand [2].

For most cereal crops, yield relates to grain production. Increasing grain number and weight are two important paths to increase yield per plant [10,11,14]. Grain number is determined during inflorescence development [12,13,14,15], and mutants and genes affecting this process have already been well studied in crops such as rice and maize [8,12,13,14,15]. Recent studies suggest that we can create new beneficial alleles by genome editing to optimize the expression or function of these genes. Indeed, some of these new alleles may have a larger effect than natural alleles [16,17,18,19,20]. These findings suggest that applying the knowledge of crop inflorescence development can help to engineer crop yield improvement, and usher in a new era of breeding practice. In this review, we focus on crop inflorescence development in rice, wheat and maize (Figure 1), and how to use this knowledge to improve yield by increasing grain number. We also touch upon emerging technologies to efficiently genome edit diverse germplasm.

## 2. Grain Number Is Determined during Inflorescence Development in Cereal Crops

Grains or kernels grow on the panicle in rice, and the spike in wheat, barley and maize. These are the two typical inflorescence architectures in cereal crops [21,22], and they differ in branching architecture, as the panicle has both long and short branches, while the spike has only short branches [21,22]. Unlike rice and wheat, modern maize produces two distinct inflorescences, the tassel and ear [15]. The tassel bears staminate flowers and is borne at the apex of the mature plant, whereas the ear bears pistillate flowers to form kernels [15]. In this review, we only discuss the ear inflorescence in maize. The inflorescence stem is called the rachis, with a series of nodes that produce additional long branches or short branches (spikelets) [21]. In the rice panicle, the rachis can produce primary and secondary branches in a spiral phyllotaxy and bears single spikelets [21,22]. The rachis produces only a series of short branches in the wheat and maize spike, called spikelets [14,15]. However, the panicle and spike have a similar fundamental development. After the transition from the vegetative phase to the reproductive phase, the vegetative shoot apical meristem (SAM) transitions into the inflorescence meristem (IM) in rice and wheat, and the axillary meristems in leaf axils form the ear IM in maize [14,15]. Each IM will further proliferate to initiate branch meristems (BM) and spikelet meristems (SM) in rice, but only SMs in wheat [14,15]. In maize, the ear IM initiates spikelet pair meristems (SPM) and each SPM forms two SMs [14,15]. Each SM will further develop into floret meristems (FM) that give rise to grains after fertilization [14,15]. Therefore, the total number of SMs generated by the IM determines the number of grains on each panicle and spike, suggesting IM activity is critical for the grain number and yield (Figure 1).

The strategies for increasing grain numbers on panicles and spikes may differ depending on the species. As the panicle has long branches, an increase in grain number could be achieved by promoting the IM to develop more BM and/or SM. In contrast, increasing grain number on cereals that make a spike can only be achieved by promoting SM development. Knowledge of the genetic regulation of inflorescence architecture can help understand the basic developmental mechanisms to optimize cereal inflorescence architecture. Studies of mutants and quantitative genetic analyses have already characterized many genes and pathways that play a critical role in inflorescence development [7,8,9,10,11,12,13,14,15]. The conserved function of critical genes across species can facilitate understanding of inflorescence development, and guide yield enhancement in other agronomically important cereal species, such as orphan crops [14]. In the following section, we introduce some essential genes and pathways that regulate inflorescence development across rice, maize and wheat, focusing on regulation of IM activity and axillary meristem formation (Figure 1).

## 3. CLAVATA–WUSCHEL (CLV–WUS) Negative Feedback Loop Maintains IM Activity

Meristem size and the number of branches relies on IM activity, which is maintained by the *CLAVATA* (*CLV*)–*WUSCHEL* (*WUS*) feedback signaling pathway, which was first discovered in *Arabidopsis* [23]. This pathway is functionally conserved in eudicots and in grasses, such as rice and maize. In this pathway, a homeodomain transcription factor, WUS, is expressed in the meristem organizing center and coordinates meristem activity by activating expression of the secreted peptide CLV3, which binds its receptor, CLV1 to repress *WUS* expression [24,25,26,27,28,29]. *CLV* orthologs in maize include the *CLV1* ortholog *THICK TASSEL DWARF1* (*TD1*) [30], and *CLV3* ortholog *ZmCLAVATA3*/*EMBRYO SURROUNDING REGION-RELATED7* (*ZmCLE7*) [31,32], as well as a second CLE peptide, *ZmFON2-LIKE CLE Protein1* (*ZmFCP1*) [33], which acts in a distinct *CLV* pathway to repress *WUS* expression. Besides the CLV1 receptor, another LRR receptor-like protein FASCIATED EAR3 (FEA3) represses *WUS* from below a region of the meristem known as the organizing center, by perceiving ZmFCP1 [33]. The *CLV2* ortholog in maize, FASCIATED EAR2 (FEA2), transmits signals from ZmCLE7 and ZmFCP1 through two different candidate downstream effectors, the maize heterotrimeric G proteins, COMPACT PLANT2 (CT2) and ZmGB1, and CORYNE (ZmCRN) [34,35,36,37]. Null mutants of these *CLV* genes cause meristem over-proliferation, leading to enlarged inflorescence stems and fasciated ears in maize [30,31,32,33,34,35,36,37].

The orthologs of *WUS*, *CLV1*, and *CLV3* in rice have also been identified. *tillers absent1* (*tab1*) is a mutant in the *WUS* ortholog in rice, and has a flat IM, and shorter rachis branches with fewer spikelets, suggesting a role in axillary meristem initiation [37]. In contrast, the IM of rice *clv1* ortholog mutant, *floral organ number1* (*fon1*), is larger and produces more long branches [38]. Similarly, *fon4*, a mutant in the *CLV3* ortholog, has a more obvious enlargement in IM size and gives rise to more than one primary rachis [39]. A QTL for grain yield and nitrogen use efficiency was identified as a rice heterotrimeric G protein subunit, *DENSE AND ERECT PANICLE1* (*DEP1*) [40,41]. A gain-of-function mutant of *DEP1* has a larger IM, and produces more long branches by enhancing the expression of *CYTOKININ OXIDASE 2* (*OsCKX2*) [40]. However, it is unclear whether DEP1 participates in the CLV–WUS signaling pathway in a manner similar to maize heterotrimeric G proteins.

The CLV–WUS signaling pathway has not been well studied in wheat. However, the ortholog of rice heterotrimeric G-protein subunit *DEP1*, *TaDEP1,* was found to act as a negative regulator of IM activity to affect spike length and spikelet number [40]. As the CLV–WUS signaling pathway function is conserved across many plant species, it is expected to have a similar role in wheat.

## 4. The Regulation of Lateral Meristem Initiation

After the transition from the vegetative meristem to the IM, the IM will proliferate to generate daughter stem cells that form new meristems called axillary or lateral meristems [14,15,21]. In a panicle, two types of lateral meristems, BM and SM, are formed and develop into long and short branches, respectively [14,15,21], whereas, only SMs are generated on a spike [14,15,21]. Therefore, axillary lateral meristems initiate the development of long branches and spikelets. In rice and wheat, inflorescence branching usually correlates with vegetative shoot branching, also called tillering. Tillering is suppressed in domesticated maize, and was achieved through selection for a gain-of-function allele of the TCP (TEOSINTE BRANCHED1, CYCLOIDEA, PCF1) transcription factor *TEOSINTE BRANCHED1* (*TB1*) to increase apical dominance compared to its wild ancestor teosinte [14,15,21,42]. A comprehensive study by ChIP-seq, RNA-seq, and hormone and sugar measurements reveals that TB1 may regulate phytohormone pathways such as gibberellins, abscisic acid and jasmonic acid, as well as sugar metabolites for energy balance [43]. *TB1* orthologs in rice and wheat function similarly to negatively regulate both tillering and inflorescence branching [44,45].

In rice, *OsTB1* is directly regulated by *IDEAL PLANT ARCHITECTURE1* (*IPA1*), a SQUAMOSA PROMOTER BINDING PROTEIN-BOX-LIKE (SPL) transcription factor (*OsSPL14*), which is critical in regulating rice vegetative and inflorescence architecture, and substantially enhances grain yield [46,47,48]. *IPA1* regulation involves miRNAs miR156 and miR529 [46,47]. A beneficial *IPA1* allele has a point mutation in the miR156 target site, perturbing regulation by this miRNA leading to upregulation of *OsSPL14* [46]. This upregulation suppresses tillering and increases the number of vascular bundles, which may contribute to water/nutrient transport and lodging resistance, and an increase in grain yield [46]. *IPA1* acts as a direct downstream component of DWARF53 (D53) signaling to affect strigolactone (SL)-induced gene expression [49]. D53 is a key repressor of the SL signaling pathway, and strigolactones are mobile root-to-shoot phytohormones that suppress shoot branching by inhibiting the outgrowth of axillary buds [48]. D53 and IPA1 proteins interact to suppress the transcriptional activation activity of IPA1 [49]. Meanwhile, IPA1 binds directly to the *D53* promoter and functions in feedback regulation of SL-induced *D53* expression [49]. As well as *OsSPL14*, *OsSPL7* and *OsSPL17* are also negative regulators of tillering, but positive regulators of the spikelet transition [50]. Fine-tuning the expression of these *OsSPLs* by miR156 and miR529 can optimize panicle size and yield [50].

The *OsSPL14* orthologs in maize, *UNBRANCHED2* (*UB2*) and *UB3*, redundantly limit the rate of cell differentiation in the lateral domains of meristems [51]. Remarkably, a kernel row number (KRN) QTL, *KRN4*, maps to a ∼3 Kb intergenic region about 60 Kb downstream from *UB3*, and interacts with the *UB3* promoter to quantitatively enhance *UB3* expression and KRN [52,53]. This chromatin-based interaction of *KRN4* with the *UB3* promoter is mediated by UB2 [53]. Thus, *UB3* negatively regulates IM size and spikelet number in maize [51,52,53]. Overexpression of maize *UB3* in rice has a dosage-dependent effect on panicle branch number and grain yield [54]; *UB3* expression promotes grain yield, but high expression suppresses plant growth and reduces yield [54].

Consistent with its function in rice, overexpression of miR156 in wheat leads to increased tiller number and severe defects in spikelet formation, probably due to repression of a group of *SPL* genes [55]. Strigolactone signaling repressor TaD53 also directly interacts with the N-terminal domains of TaSPL3/17, suggesting association between miR156-TaSPLs and SL signaling pathways during wheat tillering and spikelet development similar to in rice [55].

The SMs are paired in maize, and single in wheat and rice [15]. However, a mutation in *TaTB1* converts single to paired spikelets in modern bread wheat cultivars, and increases spikelet number [45]. Variation in spikelet row-type is also studied in barley, which is in the same *Triticeae* tribe and has an unbranched spike inflorescence similar to wheat. Each rachis internode in barley develops one central and two lateral spikelets [56]. In two-rowed varieties, the central spikelet is fertile and produces grain, and the two lateral spikelets are sterile [56]. The *TB1* ortholog in barley, *INTERMEDIUM-C* (*INT-C*), controls lateral spikelet fertility, resulting in changes from two-rowed to six-rowed varieties, where all three spikelets are fertile and develop into grains [57]. The six-rowed phenotype is controlled by several additional loci, including *SIX*-*ROWED SPIKE1* (*VRS1*) [58], *VRS2* [59], *VRS3* [60], and *VRS4* [61]. *VRS3*, *VRS4*, and *INT-C* act as transcriptional activators of *VRS1* and control the number of fertile lateral spikelets [62]. The orthologs of *VRS1* and *VRS4* in maize are *GRASSY TILLERS1* [63,64] and *RAMOSA2* [65], which act as negative regulators of shoot and ear branching, respectively. As *TB1* orthologs have a similar function in grain row formation in wheat and barley, manipulating these *VRS* orthologs may also increase the grain number in wheat.

## 5. Optimizing Inflorescence-Development-Related Genes to Enhance Crop-Yield Traits

Targeted genome editing is a powerful and simple tool that allows new possibilities to modify genomic sequences, accelerating gene function analysis and speeding up breeding by creating favorable alleles. The improvements in genome editing tools and transformation methods help researchers overcome the challenges of complex genomes and lack of mutants due to genetic redundancy in cereal crops. In particular, the application of clustered regularly interspaced short palindromic repeats (CRISPR)-Cas family-based technology allows the generation of mutations in almost any gene of interest. It is also possible to fine-tune function of a target gene, for example, editing key inflorescence development regulators can produce new high-yielding alleles. Next, we introduce four cases that have implemented genome editing to optimize the function of inflorescence-related genes to enhance grain yield.

### 5.1. Case 1: DEP1 and IPA1 Coding Sequence Mutagenesis by CRISPR-Cas9 to Enhance Grain-Yield-Related Traits in Rice

Mutants in *DEP1* increase panicle size and grain number [40], and mutations in the miR156 cleavage site of *IPA1* have “ideal plant architecture”, such as fewer tillers, more grains per panicle, and sturdy stems, substantially enhancing rice grain yield [46]. Li et al. used CRISPR-Cas9 to knockout the *DEP1* coding region and mutate the miR156 cleavage site in *IPA1* in a single rice cultivar [66]. Two *DEP1* frameshift mutations resulted in shorter plants and panicles, with more flowers per panicle [66]. An *ipa1* CRISPR allele, interrupting the miR156 cleavage site, led to a decrease in tillers and increase in plant height, flower number and panicle length, similar to a previously characterized allele with a cleavage site mutation [46,66]. These results suggest that CRISPR-Cas9 can edit key regulators of important traits to modify these traits in cultivated rice varieties [66].

### 5.2. Case 2: Producing Beneficial Promoter-Edited Alleles of OsTB1 by CRISPR-Cas9 to Enhance Grain-Yield-Related Traits in Rice

Most ‘super rice’ varieties with high yields have several beneficial agronomic traits, including strong culms for lodging resistance and large panicles for high yield [20]. Cui and Hu et al. used a chromosome segment substitution population to map culm strength, by measuring the stem cross-section area (SCSA) of the fourth internode, and found that *OsTB1* was the causative gene of a major QTL [20]. A TGTG insertion in the 5′ UTR was predicted to control *OsTB1* expression, and consequently SCSA [20]. Next, the authors introduced mutations in the promoter and 5′-UTR of *OsTB1* by CRISPR-Cas9 using six sgRNAs [20]. Nine different mutations were obtained, including large and small deletions and by chance one that recreated the TGTG insertion. These mutations were classified into three types, based on the effect on expression and phenotype. Type 1 alleles had large deletions covering an *OsSPL14* binding site and part of the 5′-UTR, and these lines had lower *OsTB1* expression and more tillers with smaller culms and panicles [20]. Type 2 alleles had changes in other parts of the *OsTB1* promoter, and had no change in gene expression or phenotype [20]. Type 3 alleles had higher gene expression and more tillers with larger culms and panicles, and one of them had the TGTG insertion, supporting its role as a QTL [20]. Therefore, CRISPR-Cas9 could produce desirable alleles with appropriate levels of expression for optimizing breeding targets [20].

### 5.3. Case 3: Promoter Editing of CLE Genes and Knockout of a Redundant Paralog by CRISPR-Cas9 to Enhance Grain-Yield-Related Traits in Maize

Null alleles of maize *clv* genes cause meristem over-proliferation and fasciated ears that develop many more disorganized and shorter kernel rows with low grain yield [30,31,32,33,34,35,36,37]. However, weak coding sequence alleles of *fea2* and *fea3* generated by ethyl methanesulfonate mutagenesis cause a quantitative increase in kernel row number while maintaining meristem organization and ear length, highlighting the potential to quantitatively manipulate *fea* genes for yield enhancement [33,37]. This idea was tested by Rodríguez-Leal et al., who generated weak alleles by CRISPR–Cas9 editing of cis-regulatory regions, such as promoters, in tomato [16]. Liu et al. tested this strategy in maize, by CRISPR–Cas9 multiplex editing of the promoters of *ZmCLE7* and *ZmFCP1.* This produced weak alleles that maintain normal ear length but with a quantitative increase in KRN and grain yield [17]. Potential regulatory regions for editing were predicted using chromatin state (assay for transposase accessible chromatin with sequencing and micrococcal nuclease digestion with sequencing) and conserved non-coding sequence (CNS) data [17]. Promoter-edited alleles with deletions in these regions had lower *ZmCLE7* and *ZmFCP1* expression, leading to an increase in IM size, kernel number and grain yield [17]. In contrast to these deletion alleles, one allele in *ZmCLE7* carrying an inversion produced opposite effects, decreasing kernel number and grain yield, possibly due to an expansion of *ZmCLE7* expression and a decrease in meristem size and yield-related traits [17].

In addition to weak allele promoter editing, Liu et al. also explored redundant compensators of *ZmCLE7* through transcriptome analysis of *Zmcle7* mutants. They found a previously uncharacterized maize CLE gene, *ZmCLE1E5*, that was upregulated in *Zmcle7* mutants [17,31]. CRISPR–Cas9 edited null alleles of *ZmCLE1E5* had much weaker meristem size phenotypes than *Zmcle7*, but significantly enhanced it [17]. The *Zmcle1e5* null alleles also quantitatively enhanced grain-yield-related traits, such as KRN and grain yield per ear [17]. The three CLE genes edited in this study had not been previously linked to yield-related QTLs in maize diversity populations, and *ZmCLE7* regulatory regions have low genetic variation [17]. Thus, CRISPR–Cas9 genome editing produced new beneficial maize alleles, even for genes not previously associated with yield traits, and had effects greater than molecularly characterized QTLs [17]. However, these studies used lab strains of maize, and it remains to be tested how such alleles will perform in elite varieties.

### 5.4. Case 4: Uncovering Conserved Gene Functions and Engineering Quantitative Trait Variation by CRISPR-Cas9 Cis-Regulatory Mutations in the Tomato CLV–WUS Pathway

Besides the random-promoters mutagenesis by CRISPR to screen beneficial alleles in cereal crops [17,20], two studies conducted precise editing of predicted cis-regulatory sequences in tomato. Loss of tomato *SlCLV3* results in enlarged meristems that cause fasciated phenotypes. Wang et al. used CRISPR to precisely edit the CNSs in *SlCLV3* and *SlWUS* promoters and studied the functional relationships of cis-regulatory sequences in their promoters [18]. Two conserved CNSs were found that contained functional sequences and showed additive, synergistic, and redundant relationships to contribute to *SlCLV3* promoter function [19]. Unlike the *SlCLV3* promoter, the *SlWUS* promoter is more tolerant to perturbations, as most of its promoter-edited mutants appeared normal [18,67]. Besides *cis*-regulatory interactions, Hendelman et al. revealed that the conserved cis-regulatory sequences of *WOX9* are responsible for its pleiotropic activity in embryo and inflorescence development [19,68]. The authors used CRISPR to edit these conserved cis-regulatory sequences and obtained a comprehensive promoter-edited allelic series to expose multiple pleiotropic roles of SlWOX9 [19]. Promoter deletions in two separate regions were tightly associated with branched inflorescences and embryonic lethality phenotypes, respectively [19]. These findings suggested precise cis-regulatory region mutagenesis by genome editing can discover hidden conserved pleiotropy [19], which is important for generating beneficial alleles when a negative pleiotropic effect on other traits needs to be limited. Therefore, genome editing tools can be used to dissect the complex *cis*-regulatory interactions, pleiotropic functions and engineer variants to shape traits quantitatively, which can guide the precise design of cis-regulatory alleles for cereal crop improvement.

## 6. Challenges and Emerging Technologies for CRISPR/Cas9-Based Crop Improvement

After just a little under a decade, CRISPR/Cas9 has revolutionized gene editing capabilities in a wide range of organisms, including cereal crops [69,70]. The case studies in the previous section demonstrate the power of CRISPR/Cas9 to modulate gene activity to improve crop yield. Still, several milestones must be reached to make CRISPR a routine tool for crop improvement, including engineering precise edits, stacking mutations, and introducing edits into agriculturally important germplasm. Here, we highlight several recent advances in CRISPR technologies and techniques with a promise to more efficiently deploy gene editing in cereal crops (Figure 2).

### 6.1. CRISPR Techniques

CRISPR/Cas9 uses the Cas9 enzyme to generate double-stranded breaks in a specific DNA sequence guided by sgRNA molecules that recognize a genomic region of interest. Double-stranded breaks are then repaired via either non-homologous end joining (NHEJ) or homologous recombination (HR). NHEJ is error prone, producing small insertions and deletions in the DNA sequence, generating mutations in regions of interest. Larger deletions and insertions can also be generated at a lower frequency. Most plant systems rely on *Agrobacterium*-mediated or biolistic delivery of a plasmid containing Cas9 under the expression of a constitutive promoter and the sgRNA sequences driven by PolIII promoters. An extensive description of CRISPR editing in monocots has recently been reviewed elsewhere [71,72].

To create single base-pair substitutions in a genomic region of interest, for example to change a coding sequence, CRISPR base editing can be used [73]. In this technique, a catalytically inactive Cas9 (dCas9) is coupled to a deaminase to convert one base pair to another. Since this method uses a dead Cas9, the introduction of double-strand breaks and homology-directed repair do not occur, which allows for more precise editing. CRISPR base editing has been successfully applied in many plants, including maize, rice, and wheat [74,75,76,77].

Generally, CRISPR target sequences are limited to the NGG protospacer adjacent motif (PAM) when using the Cas9 enzyme. While it is generally possible to select targets with the NGG motif to edit genomic regions of interest, there are instances when a very specific target sequence is desired. Recent work from Ren et al. and Xu et al. relaxed the requirement for a specific PAM sequence, allowing for greater control over choice of region for CRISPR editing [78,79]. This technology, called SpRY, uses a version of Cas9 that has 11 amino acid substitutions to enable CRISPR targeting of any site (NNN), and has been successfully used to edit rice and conifers. A SpRY base editor was also developed and could also edit relaxed target sites. It should be noted, however, that edits in the original sgRNAs transformed into the plants were observed for SpRY, likely a result of the guides editing themselves, which could result in off-target effects.

While CRISPR mutations derived from non-homologous end joining have been widely reported in plants, creating targeted insertions via homologous recombination remains a challenge. However, a more efficient method to select for precise gene insertions into plant genomes has been recently reported. In this method, Cas9 is expressed early during maize transformation, and mobilizes a donor template flanked with HR sites. Excision of the donor template by Cas9 activates a selectable marker gene, resulting in the selection only of transformants that have successfully undergone homologous recombination [80]. Another study used CRISPR/Cas9 to create a site-specific landing pad to insert multiple transgenes into selected sites in the maize genome [81]. Other non-CRISPR-based methods are being developed to precisely deliver transgenes into plants, such as recombineering [82].

An additional method to overcome the imprecise nature of edits derived from NHEJ has been developed [83,84]. With this technology, called prime editing, a catalytically impaired Cas9–nickase is fused to a reverse transcriptase (RT). A prime-editing guide RNA (pegRNA) guides the Cas9–RT fusion, and encodes the desired DNA changes. This system can produce precise base-pair changes, as well as small transversions, insertions and deletions in wheat and rice. While initial reports of efficiency for prime editing in plants were low, it was substantially improved by modifying the pegRNA melting temperature [85].

As demonstrated with CRISPR editing of cis-regulatory regions, modulating gene expression can have a beneficial impact on yield. As an alternative to editing the DNA of cis-regulatory regions, CRISPR activation or inactivation can be employed to alter gene expression. This technique also uses dCas9, which is fused with different protein domains to regulate gene expression. For example, the SunTag system can be used to activate gene expression when combined with the transcriptional activator VP64, as well as alter methylation patterns when combined with methylation or demethylation effectors [86,87,88]. In this system, dCas9 is fused to tandem GCN4 epitopes, bringing them to a specific locus, where they are detected by a GCN4 antibody fused with either the VP64 activator or methylation enzymes. In the cases of the dCas9 fusions with methylation enzymes, induced epigenetic changes were heritable over many generations in the absence of the transgene in *Arabidopsis*. This suggests the approach could be used to induce changes in expression in crop plants in a non-GMO context. Other protein domains fused to dCas9 have been reported to have even higher transcriptional activation capabilities than VP64, but the degree of activation is gene-dependent, so more work will be needed to understand the underlying mechanisms [89,90].

Beyond modifying Cas9 to generate specific types of edits, predicting the on-target efficiency and off-target effects of particular sgRNAs remains challenging. Recently, however, machine learning approaches have been developed to more efficiently predict these two parameters, and can reduce the guesswork required when designing efficient sgRNAs [91,92].

### 6.2. Increasing Transformation Efficiency in Cereals

Efficient *Agrobacterium*-mediated or biolistic transformation of cereal crops is restricted to specific species and cultivars [93,94,95]. As a result, it is essential to develop new technologies to increase transformation efficiency for CRISPR mutagenesis of recalcitrant crop germplasm.

One effort to improve transformation efficiency has involved ectopically expressing developmental (DEV) genes necessary for embryogenesis and meristem formation, including those encoding the maize embryonic transcription factor BABYBOOM (BBM) and meristematic transcription factor ZmWUSCHEL2 (ZmWUS2) to improve transgenic tissue regeneration [96,97,98]. *Agrobacterium*-mediated delivery of a cassette driving overexpression of both of these genes into recalcitrant lines of maize, rice, sugarcane, and sorghum resulted in increased transformation efficiency. These methods have expanded the number of maize varieties amenable to transformation, with varying degrees of success, but expression of the DEV genes must be tightly regulated, because ectopic expression results in growth defects.

A more recent advance in plant transformation uses a transcription factor chimera, consisting of wheat meristem regulators GROWTH-REGULATING FACTOR 4 and GRF-INTERACTING 1 (GRF–GIF), to increase both regeneration efficiency and speed [99]. This has resulted in a more efficient transformation of wheat (up to approximately 80%), rice, and triticale. It has also expanded the number of wheat cultivars amenable to transformation, including bread and durum wheat lines. Overexpression of maize *GRF5*, a homolog of wheat *GRF1*, also increased transformation efficiency in the maize A188 line, suggesting the GRF–GIF chimera may further increase transformation efficiency in this crop plant [100]. The efficiency of the GRF–GIF system appears to be limited to specific genes within the GRF family, as a higher transformation efficiency was reported in maize when using maize GRF5, but not the *Arabidopsis* ortholog. Furthermore, while the wheat GRF–GIF chimera could successfully increase rice transformation efficiency, this system may need to be fine-tuned on a per species basis to achieve the same boosts seen in wheat transformation. Debernardi et al. postulate that because GRF–GIF and BBM–WUS act at different stages of meristem differentiation and proliferation, the two technologies could be combined to synergistically enhance overall transformation efficiency.

A third transformation strategy is to deliver CRISPR transgenes into de novo induced meristematic tissue, instead of transforming callus generated from explants [101]. In one such study, axillary shoot meristems were removed from Cas9-expressing *N. benthamiana* plants, which were then treated with *Agrobacterium* strains expressing developmental regulators, a luciferase reporter to assess transformation efficiency, and a sgRNA targeting *PHYTOENE DESATURASE* to allow visual assessment of CRISPR-editing efficiency. Ectopic expression of the developmental regulators *SHOOT MERISTEMLESS* (*STM)* and *ZmWUS2*, or *ZmWUS2* and *Agrobacterium* tumor-inducing gene *isopentenyl transferase (ipt)*, produced de novo meristems, which eventually formed flowers that produced seeds with edits generated from the sgRNA. So far, this strategy has only been successful in dicots, but if it can be extended to cereals, it will greatly accelerate the production of CRISPR mutations. Indeed, particle bombardment of wheat shoot meristems can produce transformants, indicating that this technique could be applied to cereals [102].

### 6.3. Use of Viral Vectors for CRISPR Mutagenesis

In addition to improving existing plant-transformation technologies, viral vectors are being explored as an alternative to deliver CRISPR reagents into plants. When *Cas9* transgenic plants were infected with RNA viruses engineered with sgRNAs tagged with the Flowering Locus T (FT) mobile RNA element, they successfully edited germline cells at high frequency [103]. Thus, with a single transgenic line, multiple guide RNAs can be delivered, greatly reducing the number of stable transformants needed for CRISPR. Viral vectors are generally limited by insert size, but a plant RNA rhabdovirus was engineered to harbor both Cas9 and sgRNAs and successfully edited *Nicotiana benthamiana* plants [104]. While so far demonstrated only in this species, these methods may be extended for cereal crop gene editing. For example, the foxtail mosaic virus (FoMV) was used to transiently express transgenes and deliver a sgRNA into *Setaria viridis*, *Nicotiana benthamiana*, and maize [105]. Future work should explore the most appropriate viral vectors and mobile RNA elements to maximize editing efficiency in germline tissues of cereals for heritable mutations.

### 6.4. Haploid Induction Plus CRISPR to Introgress into Elite Varieties

Introgressing a favorable allele into a commercial maize inbred line requires at least six generations of backcrossing to recreate the inbred background. Haploid induction followed by chromosomal doubling is a routine technique to rapidly generate inbred lines for use in developing hybrid maize varieties [106]. Haploids can be generated by crossing plants using pollen from *matrilineal/not like dad/ZmphospholipaseA1* (*mtl/nld/Zmpla1*) mutants via a poorly understood mechanism [107,108,109]. Work from Kelliher et al. demonstrated the efficacy of combining CRISPR with the *mtl* haploid inducer in maize, editing the yield-related genes *GRAIN WEIGHT 2 (GW2) GW2-1* and *GW2-2*, as well as *VRS1-LIKE HOMEOBOX PROTEIN (VLHP1)* and *VLHP2*, in several elite lines of maize [110]. Another group successfully edited *LIGULELESS1* (*LG1)* and *UB2* in the maize B73 inbred line with a related haploid inducer line containing CRISPR/Cas9 [111]. Remarkably, durum wheat was also edited by using interspecies crosses of haploid inducing maize pollen carrying CRISPR/Cas9 onto wheat [112]. As an alternative to *mtl*-based haploid induction, *centromeric histone H3* (*cenh3*) heterozygous maize mutants can also induce haploids at levels up to 20%, so this system could also be combined with CRISPR/Cas9 to quickly produce edited plants in a wide array of germplasm and bypass transformation [113]. Haploid induction strategies assisted with seed-specific marker lines will help improve the efficiency of identifying CRISPR-edited plants with this technique to be able to rapidly identify haploids at the seed stage [114,115].

Here we have highlighted just a few exciting technological advances in delivering CRISPR gene editing to crop plants. Other emerging methods include nanotechnology-based delivery of CRISPR reagents, transient expression of transgenes by spraying viral or *Agrobacterium* nanoparticles, and machine learning algorithms for improving stable transformation [116,117,118,119,120,121]. Combining several of these methods will likely allow for increased editing and introgression of CRISPR alleles into cereal crops, including agriculturally important germplasm.

## 7. Future Prospects

Most crop genome editing and evaluation of yield-related traits has not been in commercial elite lines [17,20]. However, a weak *CLV* allele was recently found to increase yield up to 28% in elite hybrids in Vietnam [122]. This suggests great potential for application of genome-edited alleles in commercial varieties. A recent study integrated multiplexed CRISPR/Cas9-based high-throughput targeted mutagenesis with genetic mapping and genomic approaches to successfully target 743 candidate genes, indicating that genome editing has advanced to a high-throughput stage [123]. This review focused on knowledge of inflorescence yield-related traits in cereal crops, including key developmental regulatory genes. It is becoming clear that quantitative variation of yield-related traits can be successfully engineered by editing of developmental genes, suggesting that crop breeding does not need to depend on naturally occurring mutations. Instead, artificially generated variation can be the raw material for future breeding. With advances in genome-editing technologies and the increasing understanding of biological traits, knowledge-driven breeding by design is becoming a reality that can be deployed to improve the grain yield of current elite inbred lines and to domesticate wild ancestors *de novo* to recreate new crop varieties [124]. These versatile technologies will allow us to generate diverse germplasm for a growing population and engineer more resilient crops under changing climate conditions.

## Figures and Tables

**Figure 1 ijms-22-05167-f001:**
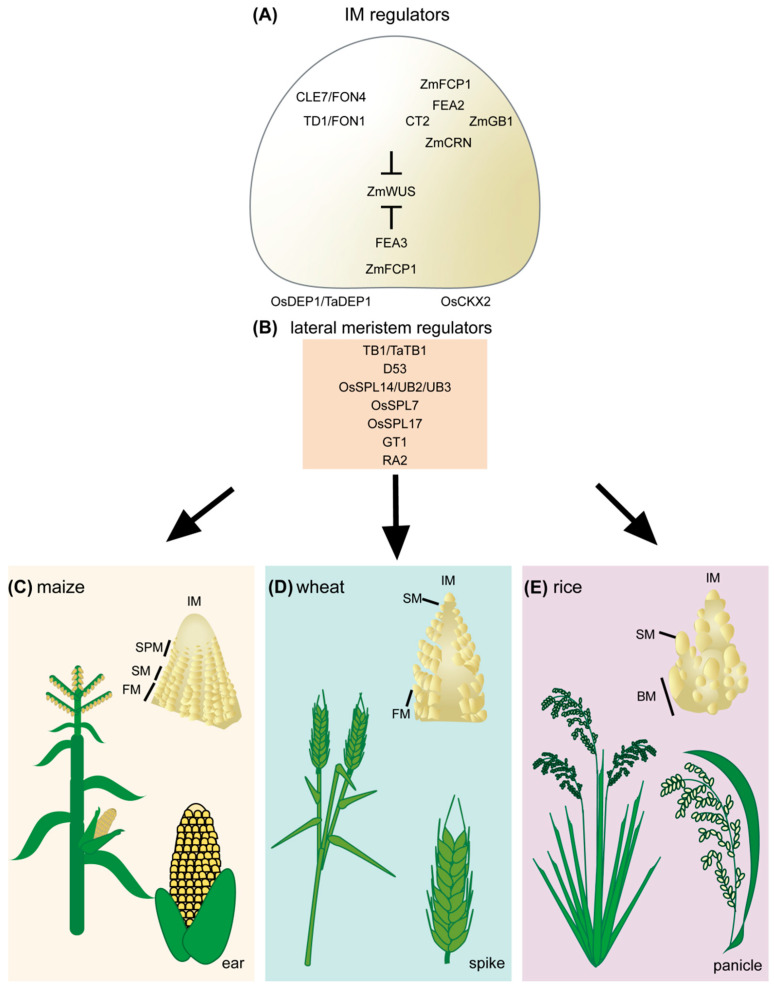
Regulators of cereal inflorescence development. (**A**) Inflorescence meristem regulators include the receptor and receptor-like proteins TD1/FON1, FEA2, and FEA3. These proteins perceive secreted CLE peptides, including ZmCLE7/FON4 and ZmFCP1. Perception of CLE peptides restricts ZmWUS activity. Downstream signaling components of FEA2 include CT2, ZmCRN, and ZmGB1. The rice DEP1 G protein also contributes to inflorescence meristem regulation through OsCKX2, though it is unclear if it acts in the CLV signaling pathway. (**B**) Lateral meristem regulators include TB1/TaTB1, D53, OsSPL14/UB2/UB3, OsSPL7, OsSPL17, GT1, and RA2. Inflorescence development and architecture of maize (**C**), wheat (**D**), and rice (**E**). IM, inflorescence meristem; SPM, spikelet pair meristem; SM, spikelet meristem; FM, floral meristem; BM, branch meristem.

**Figure 2 ijms-22-05167-f002:**
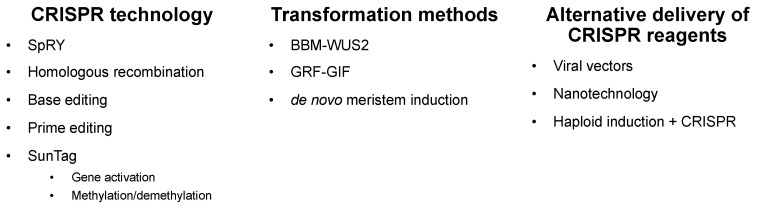
Recent technological advances to facilitate CRISPR/Cas9 editing in cereals. Emerging CRISPR technologies include SpRY, a mutated form of Cas9 with a relaxed PAM requirement; CRISPR base-editing, in which individual base pairs are changed; prime editing, in which a Cas9–nickase is fused to a reverse transcriptase to produce precise knock-in mutations guided by a pegRNA template; homologous recombination, where larger insertions can be introduced into a genomic region of interest; and SunTag gene activation and methylation/demethylation, where a catalytically inactive Cas9 is fused to effectors that modulate gene expression or methylation patterns. CRISPR technology can be delivered into cereals, using improved transformation methods involving developmentally important genes, such as *BABYBOOM–WUSCHEL* or *GROWTH-REGULATING FACTOR 4* and GRF-INTERACTING 1 (*GRF–GIF*), or through de novo meristem induction. In addition to using stable transformation to enable CRISPR editing, alternative delivery of CRISPR reagents includes the use of viral vectors, nanotechnology, and haploid induction plus CRISPR.

## Data Availability

The review did not report any data.

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
