# Peer review of "Next Generation Cereal Crop Yield Enhancement: From Knowledge of Inflorescence Development to Practical Engineering by Genome Editing"

_ijms, 2021, doi:10.3390/ijms22105167_

Round 1

Reviewer 1 Report

This review focuses on current knowledge concerning inflorescence yield-related traits in cereal crops, emerging genome editing technologies, and recent research that use inflorescence development knowledge to engineer crop yield improvement through genome editing, bringing a new era of breeding practice.

  • Abstract is less information. Please improve it.
  • In Conclusion, the authors should add the significance of this research, and potential practical application.

Author Response

Reviewer 1:
This review focuses on current knowledge concerning inflorescence yield-related traits in cereal crops, emerging genome editing technologies, and recent research that use inflorescence development knowledge to engineer crop yield improvement through genome editing, bringing a new era of breeding practice.

-Abstract is less information. Please improve it.

We thank the reviewer for this suggestion. We have revised the abstract to add more information as follows:

Artificial domestication and improvement of the majority of crops began approximately 10,000 years ago in different parts of the world to achieve high productivity, good quality, and widespread adaptability. It was initiated from a phenotype-based selection by local farmers, and developed to current biotechnology-based breeding to feed over 7 billion people. For most cereal crops, yield relates to grain production, which could be enhanced by increasing number and weight. Grain number is typically determined during inflorescence development. Many mutants and genes for inflorescence development have already been characterized in cereal crops. Therefore, optimization of such genes could fine-turn yield-related traits, such as grain number. With the rapidly advancing genome-editing technologies and understanding of yield-related traits, knowledge-driven breeding by design is becoming a reality. This review introduces knowledge about inflorescence yield-related traits in cereal crops, focusing on rice, maize and wheat. Next, emerging genome editing technologies, and recent studies that apply this technology to engineer crop yield improvement by targeting inflorescence development are reviewed. These approaches promise to usher in a new era of breeding practice.

-In Conclusion, the authors should add the significance of this research, and potential practical application.
We thank the reviewer for this suggestion. We have revised the last paragraph of the manuscript to explain clearly about the significance and potential practical application of this review as follows:

Most crop genome editing and evaluation of yield-related traits has not been in commercial elite lines [17,20]. However, a weak CLV allele was recently found to increase yield up to 28% in elite hybrids in Vietnam [122]. This suggests great potential for application of genome-edited alleles in commercial varieties. A recent study integrated multiplexed CRISPR/Cas9-based high-throughput targeted mutagenesis with genetic mapping and genomic approaches to successfully target 743 candidate genes, indicating that genome editing has advanced to a high-throughput stage [123]. This review focused on knowledge of inflorescence yield-related traits in cereal crops, including their and key developmental regulatory genes. It is becoming clear that quantitative variation of yield-related traits can be successfully engineered by editing of developmental genes, suggesting that crop breeding doesn’t need to depend on naturally occurring mutations. Instead, artificially generated variation can be the raw material for future breeding. With advances in genome-editing technologies and the increasing understanding of biological traits, knowledge-driven breeding by design is becoming a reality that can be deployed to improve the grain yield of current elite inbred lines and to domesticate wild ancestors de novo to re-create new crop varieties [124]. These versatile technologies will allow us to generate diverse germplasm for a growing population and engineer more resilient crops under changing climate conditions. 

Reviewer 2 Report

It is confusing that the review targets rice, wheat, and maize and explains the genetic basis of the row-type of barley comprehensively. I would suggest the authors include studies of barley not only for the row-type or omit this part.

L215: The common terminology for CRISPR is interspaced rather than interspersed.

I would suggest to re-order the examples in the section related to the CRISPR approaches. As they start with rice, the following example is on maize and then back to rice. Why not first talk about rice and then other crops? And why the authors include tomato finally? Is this the only example outside cereals for investigations on the WUS-CLV pathway? I would omit the tomato part or give a broader view, not just focusing on the tomato.

Subheading 4.3: Here, the authors describe a CRISPR approach targeting OsTB1. Please explain the differences between the three types of mutant alleles. The reader of this review doesn’t want to get this information by reading the original manuscript.

The caption for Fig. 2 is relatively poor. What are the abbreviations SpRY, BBM, GRF-GIF stand for? What is the mode of action for the Sun Tag?

L367: As U3 promoters have been frequently used, I would suggest including or rewrite by making this more general (PolIII promoters).

I do not find the description of prime editing. As this has overcome most of the limitations of other base-editing approaches and was published in 2019 and had been shown in rice and wheat, I expect to read about it.

I summary, the review is not clearly structured and is missing main developments in the field of targeted genome editing (e.g., Lin et al., 2020; Anzalone et al., 2019).

Reviewer 3 Report

The topic is very attractive. However, there are some concerns to be considered previous to accept the manuscript:

Line 345, Section 5: The importance of the study of off-target and on-target activities should be highlighted (Please see these papers: https://doi.org/10.1007/s11103-020-01102-y; https://doi.org/10.3390/molecules26072053)

Line 410, Section 5.2: More information on Morphogenic Genes (BBM, WUS, GRF) should be presented (Please see these papers: https://doi.org/10.3390/plants8020038; https://doi.org/10.1016/j.tplants.2020.12.001)

Line 464, Section 5.4: The benefits of CRISPR-mediated haploid production over other methods such as androgenesis should be discussed (See this paper: https://doi.org/10.1038/s41477-020-0664-9).

Recently Machine learning algorithms (MLAs) are suggested as powerful tools for modeling and optimizing gene transformation and in vitro culture-based breeding methods. I suggest including MLAs as promising tools for modeling and optimizing gene transformation and in vitro culture systems at least in the conclusion of your manuscript (https://doi.org/10.1371/journal.pone.0239901; https://doi.org/10.1016/j.plantsci.2019.03.020; https://doi.org/10.1007/s00253-020-10888-2).

Round 2

Reviewer 1 Report

Requested corrections were completed.

Reviewer 2 Report

I did not find the reply to my concerns but could see that they included some of my recommendations in the revised manuscript. I am still unhappy about the tomato story, but the authors won't follow my comments. 

Reviewer 3 Report

All the comments have been addressed. This version of the manuscript can be published in IJMS.